# Epidemiology and seasonal patterns of hospitalized dengue cases in Muscat Governorate, Oman (2022–2023): A retrospective study

Ibtisam Khalifa Al-Maskari[1], Sanjay Jaju[1,¤a], Zainab M. Al-Zadjali[1], Amal Malehi[1‡], Asiya Al Hasni[3‡], Khaleefathullah A. Sheriff[4‡], Hilal Al Sidairi[4‡], Adil Said Al Wahaibi[2¤b]*

1 Department of Family Medicine and Public Health, College of Medicine and Health Sciences, Sultan Qaboos University, Muscat, Oman, 2 Directorate General for Disease Surveillance and Control, Ministry of Health, Muscat, Oman, 3 Higher Institute of Health Specialties, Ministry of Health, Muscat, Oman, 4 Department of Medical Microbiology Laboratories, The Royal Hospital, Ministry of Health, Muscat, Oman

☯ These authors contributed equally to this work.
‡ AM, AAH, KAS, and HAS also contributed equally to this work.
¤a Current address: Department of Family Medicine and Public Health, College of Medicine and Health Sciences, Sultan Qaboos University, Muscat, Sultanate of Oman
¤b Current address: Directorate General for Disease Surveillance and Control, Ministry of Health, Muscat, Oman
* adil.alwahaibi@moh.gov.om

## Abstract

This study describes the epidemiological, clinical, and seasonal characteristics of hospitalized dengue fever (DF) cases in Muscat Governorate, Oman (2022–2023) and identifies factors associated with wet-season dengue occurrence. A retrospective analytical study was conducted using national dengue surveillance data of 345 laboratory-confirmed DF patients admitted to Sultan Qaboos University Hospital (SQUH) and the Royal Hospital (RH). Sociodemographic, clinical, and laboratory data were analyzed using descriptive statistics and logistic regression to identify factors independently associated with wet-season dengue cases. The results showed that of 345 patients, 54.8% were male, and 94.5% were Omani nationals. Most admissions occurred at RH (56.8%) and during the dry season (58.3%). Adults aged 41–60 years comprised the largest age group (44.6%). A distinct seasonal pattern was observed, with peaks in April-May of both years. Seeb Wilayat accounted for 69.3% of cases, followed by Bawshar (26.1%) (p = 0.027). Clinically, 84.6% had dengue with warning signs, while 15.4% developed severe dengue. Locally acquired infections represented 98.6% of cases. Hypertension (31.9%) and diabetes mellitus (25.2%) were the most common comorbidities. PCR positivity and hospital admission were more frequent during the wet season; however, these patterns are likely to reflect differences in testing and referral practices rather than increased dengue transmission. Overall, 96.2% of patients recovered, and 3.8% (n = 13) died. Multivariable analysis

**Data availability statement:** The data that support the findings of this study are not publicly available due to ethical and legal restrictions on sharing health surveillance data imposed by the Ministry of Health, Sultanate of Oman. These restrictions are in place because the dataset derives from national notifiable disease surveillance and contains sensitive epidemiological information. The minimal anonymized dataset necessary to replicate the main findings (aggregated case counts by time, age group, gender, and nationality) is available upon reasonable request and with permission of the Ministry of Health. Requests may be directed to the corresponding author, who will coordinate with the Directorate General for Disease Surveillance and Control, Ministry of Health, Oman. Alternatively, requests can be sent to the institutional data custodian: Dr. Adil Said Al Wahaibi, Head of the Epidemiological Surveillance Department, Ministry of Health, Oman (email: adil.alwahaibi@moh.gov.om).

**Funding:** The author(s) received no specific funding for this work.

**Competing interests:** The authors have declared that no competing interests exist.

**Abbreviations:** CDC, Centres for Disease Control and Prevention; DF, Dengue Fever; DHF, Dengue Haemorrhagic Fever; DSS, Dengue Shock Syndrome; EMR, The Eastern Mediterranean Region; MOH, Ministry of Health; RH, Royal Hospital; SD, Standard Deviation; SPSS, IBM Statistical Package for the Social Sciences (SPSS); SQUH, Sultan Qaboos University Hospital; WHO, World Health Organization; NS1, Non-structural protein 1 of dengue virus; IgM, Immunoglobulin M; IgG, Immunoglobulin G; PCR, Polymerase chain reaction.

indicated that wet-season admissions were more likely to occur at SQUH (aOR = 1.72; 95% CI = 1.06–2.82; p = 0.030) and among PCR-confirmed cases (aOR = 1.38; 95% CI = 1.02–1.87; p = 0.040); these associations likely reflect hospital referral and diagnostic practices rather than increased transmission risk. In conclusion, hospitalized dengue cases in Muscat Governorate exhibited clear seasonal and geographic clustering, while associations with PCR positivity and hospital of admission reflect health-system and diagnostic patterns rather than increased dengue incidence. The predominance of locally acquired infections indicates ongoing endemic transmission. Strengthening vector surveillance, particularly in Seeb and Bawshar, and enhancing hospital preparedness prior to seasonal peaks remain important public health strategies, supported by existing literature rather than this study alone.

## Introduction

Dengue Fever (DF) is an acute febrile illness resulting from infection with any of four closely related single-stranded RNA viruses belonging to the genus Flavivirus [1–4]. It is a vector-borne disease transmitted primarily by bites of female Aedes aegypti mosquitoes [5,6], and to a lesser degree, by Aedes albopictus [7,8]. While the disease does not spread directly from person to person, it remains a major cause of illness and mortality in tropical and subtropical regions [1,4]. Although its signs and symptoms can be mild, including elevated body temperature, headache, joint and muscle discomfort, accompanied by a rash, it can escalate to critical conditions such as Dengue Hemorrhagic Fever (DHF) and Dengue Shock Syndrome (DSS), posing life-threatening risks [1,5,9]. Globally, DF has become one of the fastest-growing tropical diseases, with approximately 50 million infections reported annually and over 2.5 billion people at risk [3,10]. The geographic distribution of DF has expanded, and today, around half of the global population is in danger of being affected [11]. In the Eastern Mediterranean Region (EMR), countries such as Saudi Arabia, Yemen, and Oman have reported increasing dengue outbreaks, often linked to climatic changes and urbanization [9].

Oman's DF incidence rates have steadily increased in recent years, with the rates for 2016, 2018, 2019, and 2020 reaching approximately 30, 48, 60, 120, and 300 cases per 100,000 population, respectively [8]. Despite efforts to monitor and report cases within 24 hours, underreporting remains a concern due to the disease's complex nature and the challenge of identifying all cases [11]. Most reported cases are identified through healthcare facilities and therefore tend to represent clinically apparent infections, particularly those requiring medical attention, while milder cases may remain undetected [12]. Oman shares borders with Saudi Arabia, Yemen, and the United Arab Emirates, and has a subtropical climate characterized by hot, dry conditions with occasional rainfall and seasonal winds [13]. Despite Oman's robust healthcare infrastructure, DF remains a growing public health challenge due to endemicity in neighboring countries, along with increasing cross-border travel, tourism, and urbanization [1].

Muscat Governorate, home to approximately 1.5 million residents as of February 2025, with expatriates comprising 61%, is especially vulnerable due to high population density and rapid coastal urbanization [14]. The region's coastal *Wilayats*, Seeb, Bawshar, Muttrah, Muscat, and Qurayyat, provide favorable conditions for *Aedes aegypti* breeding [15,16]. The Muscat Governorate has a tropical climate with seasonal rainfall, primarily occurring during November-April, which supports mosquito survival [17]. Despite the country's robust healthcare system, Oman faces increasing challenges related to infectious diseases, particularly DF, which has been endemic in neighboring Saudi Arabia and Yemen for years and poses a growing risk to Oman due to factors such as tourism, climate, and urbanization [1]. It is important to highlight that Oman is a popular destination, especially during its annual festivals in Muscat and Dhofar, which attract both regional and international visitors [18].

A significant local study conducted by Al Balushi et al. (2023) tracked DF cases in Muscat Governorate through e-notification and active surveillance during two major outbreaks, one from late 2018 to early 2019, and another in early 2022. Their findings underscored the growing presence of the DENV-2 serotype and identified *Aedes aegypti* larvae in 18.5% of visited sites in specific areas like Bowsher and Seeb Wilayat, further confirming the risk of sustained transmission and consistent with findings from other local outbreaks and hospital-based studies in Oman [6,19].

In alignment with Oman's Health Vision 2050, which emphasizes the integration of research into policy [20]. Applying epidemiological principles of time, place, and person [21,22], is critical for understanding dengue dynamics. This framework allows public health authorities to target interventions, allocate resources efficiently, and strengthen outbreak preparedness. Measuring the economic burden of dengue, including healthcare costs and lost productivity, is essential for guiding public health planning and prioritizing interventions [23]. Despite increasing DF incidence, limited published data exist on the epidemiology and clinical features of DF in Oman, particularly in Muscat Governorate. This study aims to characterize the epidemiological and clinical profiles of laboratory-confirmed DF cases admitted to two tertiary hospitals in Muscat Governorate between 2022 and 2023 according to seasonality. The findings will inform public health interventions and enhance preparedness for future dengue outbreaks in Oman and the broader region.

## Materials and methods

### Study design & setting

This was a retrospective cross-sectional descriptive study based on DF surveillance data collected from the Royal Hospital (RH) and Sultan Qaboos University Hospital (SQUH), two major tertiary care centers in Muscat, Oman. Surveillance case entries were validated and re-checked against hospital medical records and laboratory reports to confirm case status and key variables. Hospitalized dengue patients were included using convenience sampling to maximize sample size, acknowledging potential selection bias inherent to retrospective hospital-based data.

### Data collection tool

A retrospective cross-sectional study was conducted using secondary data from the Directorate General for Disease Surveillance and Control, Ministry of Health, Oman, covering the period from 1 January 2022–31 December 2023. The data were received in Excel format and accessed for research purposes on 15 March 2024. The researcher developed the data collection tool based on the WHO Dengue Guideline (2009) [24]. Initial review showed that key variables were missing or incomplete, and cases were reported using mobile numbers, limiting linkage to clinical data.

Therefore, additional data were retrieved from hospital information systems after obtaining permission from the IT department at SQUH. At RH, a microbiology investigator assisted in case identification and data extraction. The researchers had access to identifiable patient information during data collection; however, all data were anonymized immediately after extraction and used solely for research purposes.

## Sampling technique

This study used a non-probability convenience sampling method that included all eligible hospitalized dengue fever (DF) cases with complete clinical and laboratory records from Sultan Qaboos University Hospital (SQUH) and the Royal Hospital (RH) during the study period. Of the 1,735 dengue cases reported in Muscat Governorate between 2022 and 2023, a total of 705 patients were admitted to SQUH and RH. After applying inclusion and exclusion criteria and excluding incomplete records, 345 laboratory-confirmed cases were included in the final analysis.

Although probability sampling was initially considered, it was not feasible due to retrospective data limitations and incomplete clinical documentation for some cases. Therefore, all available eligible cases were included to maximize sample size and reduce selection bias. While this approach may limit generalizability [25], beyond hospitalized patients, it provides a reliable representation of moderate-to-severe dengue cases requiring inpatient care in Muscat Governorate.

## Inclusion/exclusion criteria

### Inclusion criteria:

1. A laboratory-confirmed DF case*,

2. Reported in the national surveillance system, MOH-Oman,

3. The period of admission is from 1st January 2022–31st December 2023,

4. Case admitted to a government (MOH) hospital from Muscat Governorate,

### Exclusion criteria:

1. Case not laboratory confirmed as positive DF (e.g., pending laboratory results),

2. Cases managed in settings other than MOH (e.g., cases managed solely through outpatient clinics or private healthcare facilities),

(*Confirmed DF patients were those who tested positive via PCR, NS1 antigen (with or without IgM), or both IgM and IgG antibodies using capture ELISA, and exhibited symptoms according to WHO 2009 guidelines [1,24,26]).

The dry season was defined as May-October, and the wet season as November-April, reflecting the seasonal rainfall distribution in Muscat Governorate, where most precipitation occurs during winter months, and summer is extremely hot and dry [17,27].

Note: Peak dengue cases in April-May occur near the transition between seasons [17]. Cases were classified according to the WHO 2009 dengue management guidelines.

## Data analysis

Descriptive statistics (frequencies, means, SDs, medians, IQRs) were used to summarize results. Seasonal distribution was analyzed by monthly case frequency. Associations between each independent variable and dengue season (dry vs. wet) were examined using cross-tabulations and Chi-square tests. Variables with $p < 0.25$ in the bivariate analysis were considered for inclusion in the logistic regression model to ensure potentially relevant predictors were retained. Logistic regression was used to identify factors associated with seasonal hospitalization patterns, acknowledging that predictors such as hospital of admission and PCR positivity reflect health-system and diagnostic practices, not intrinsic dengue transmission.. The dependent variable was season (0 = dry, 1 = wet). Variables with $p > 0.05$ were sequentially removed using backward elimination. Odds ratios (ORs) with 95% confidence intervals (CIs) were reported for retained predictors. All analyses were conducted using IBM SPSS Statistics, version 29. A two-tailed $p < 0.05$ was considered statistically significant.

All surveillance case entries were validated against electronic medical records and laboratory reports from both hospitals (TrakCare electronic patient record (EPR) at SQUH and Al-Shifa 3+ at RH). Each case was rechecked manually by the research team to ensure accuracy of demographic, clinical, and laboratory variables, minimizing data entry errors and improving dataset reliability.

### Ethics statement

This study was approved by the Medical Research Ethics Committee, Sultan Qaboos University (MREC#3439), and the Research and Ethical Review Committee, Ministry of Health, Oman (ID: MOH/CSR/24/29131). It used anonymized secondary data from the national surveillance system, ensuring confidentiality and privacy. Consent was waived as the study involved no interventions or risk of harm, in accordance with ethical approval and Oman's data protection regulations.

## Results

This study used national surveillance data and included 345 participants from the Muscat Governorate; all recruited from SQUH (43.2%) and RH (56.8%). Regarding gender distribution, 54.8% of the participants were male, and 45.2% were female. Most patients had a positive outcome, with 96.2% (n = 332) being discharged home after recovery. However, 3.8% (n = 13) of cases resulted in death. A summary of additional participant characteristics is provided in Table 1.

As captured through the surveillance system, DF cases showed a clear seasonal pattern as demonstrated in Fig 1. The highest number of cases occurred in April and May, in both years. Fewer cases were reported in other months, with the lowest number observed in July and August in both years. This pattern reflects potential seasonal influences, such as post-monsoon conditions that favor mosquito breeding.

Regarding place of residence, Fig 2 shows the distribution of admitted DF cases across the Muscat Governorate wilayat. It was found that most participants lived in Seeb (n = 239), followed by Bawshar (n = 90), with fewer in Muttrah (n = 11), and Amirate (n = 5). Seeb Wilayat had the highest number of hospitalized cases, consistent with the hospital catchment area rather than population-level risk.

Table 2 presents the clinical characteristics, comorbidity status, disease severity classifications, presence of complications, and outcome for the 345 patients admitted with laboratory-confirmed dengue fever in the Muscat Governorate during 2022–2023. Most cases (98.6%) were locally acquired DF, with just 1.4% linked to a history to recent travel. While the majority of cases fall under dengue with warning signs (84.6%), approximately 15.4% found to have severe Dengue. In terms of diagnostic tests, NS1 antigen (50.7%) and PCR (46.1%) were the most frequently positive diagnostic tests overall. PCR positivity appeared higher in the wet season (p = 0.006); however, this likely reflects the timing of testing and hospital admission rather than increased transmission, while NS1, IgM, and IgG results showed no significant seasonal differences.

As shown in Table 3, hypertension (31.9%) and diabetes mellitus (25.2%) were the most common comorbidities among patients. The distribution of most comorbidities did not differ significantly between seasons. However, autoimmune disorders were significantly less common during the wet season compared to the dry season (0.7% vs. 4.5%; p = 0.039).

The multivariate logistic regression (Table 4) identified the hospital of admission and PCR confirmation as independent predictors of dengue cases occurring during the wet season (November-April) in Muscat Governorate, Oman (2022−2023). Patients admitted to SQUH were more likely to present during the wet season than those at RH (aOR = 1.72; 95% CI: 1.06–2.82; *p* = 0.030). PCR- confirmed cases were significantly more frequent in the wet season aOR=1.38; 95% CI: 1.02–1.87; p = 0.040). Residence in specific Wilayats was associated with lower odds of wet season in univariable analysis, but this was not significant after adjustment. Dengue severity (Dengue with warning signs: 84.6%; severe dengue: 15.4%) and autoimmune comorbidities were not independently associated with seasonal variation, although autoimmune status showed a non-significant trend toward reduced odds.

**Table 1.** Sociodemographic Characteristics of Laboratory-Confirmed Dengue Fever Cases by Season, Muscat Governorate, Oman (2022–2023).

| Characteristics | Total (n = 345) | Dry Season[a] (n = 201) | Wet Season[b] (n = 144) | P-value |
|---|---|---|---|---|
| | n (%) | n (%) | n (%) | |
| **Hospital Admission** | | | | **0.004*** |
| SQUH[c] | 149 (43.2) | 100 (49.8) | 49 (34.0) | |
| RH[d] | 196 (56.8) | 101 (50.2) | 95 (66.0) | |
| **Year of Admission** | | | | 0.478 |
| 2022 | 137 (39.7) | 83 (41.3) | 54 (37.5) | |
| 2023 | 208 (60.3) | 118 (58.7) | 90 (62.5) | |
| **Gender** | | | | 0.527 |
| Male | 189 (54.8) | 113 (56.2) | 76 (52.8) | |
| Female | 156 (45.2) | 88 (43.8) | 68 (47.2) | |
| **Age (Year)** | | | | 0.732 |
| 0–20 | 22 (6.4) | 14 (7.0) | 8 (5.6) | |
| 21–40 | 76 (22.0) | 47 (23.4) | 29 (20.1) | |
| 41–60 | 154 (44.6) | 85 (42.3) | 69 (47.9) | |
| More than 60 | 93 (27.0) | 55 (27.4) | 38 (26.4) | |
| **Marital status*** | | | | 0.935 |
| Not Disclosed | 92 (26.7) | 53 (26.4) | 39 (27.1) | |
| Ever Married | 222 (64.3) | 129 (64.2) | 93 (64.6) | |
| Never Married | 31 (9.0) | 19 (9.5) | 12 (8.3) | |
| **Employment** | | | | 0.185 |
| Not Disclosed | 166 (48.1) | 92 (45.8) | 74 (51.4) | |
| Employed | 74 (21.4) | 50 (24.9) | 24 (16.7) | |
| Unemployed | 105 (30.4) | 59 (29.4) | 46 (31.9) | |
| **Nationality** | | | | 0.356 |
| Omani | 326 (94.5) | 188 (93.5) | 138 (95.8) | |
| Non-Omani | 19 (5.5) | 13 (6.5) | 6 (4.2) | |
| **Wilayat** | | | | **0.027*** |
| Muttrah | 11 (3.2) | 5 (2.5) | 6 (4.2) | |
| Bawshar | 90 (26.1) | 42 (20.9) | 48 (33.3) | |
| Seeb | 239 (69.3) | 152 (75.6) | 87 (60.4) | |
| Amirate | 5 (1.4) | 2 (1.0) | 3 (2.1) | |

[a]Dry season: May-October. [b]Wet season: November-April. [c]SQUH: Sultan Qaboos University Hospital. [d]RH: Royal Hospital. *Significant at P ≤ 0.05.

## Discussion

This study provides a detailed overview of hospitalized Dengue Fever (DF) cases in Muscat Governorate from 2022 to 2023, highlighting their epidemiological, geographic, and clinical features, as well as factors associated with wet-season DF. A well-defined descriptive epidemiology question aims to quantify and characterize a feature of population health, specifying the target population by person and place, and time, along with the outcome and measure of occurrence [22]. Applying this framework, our study analyzed hospitalized dengue cases in Muscat Governorate in terms of temporal trends, geographic clustering, and patient characteristics, providing insight into factors associated with seasonality.

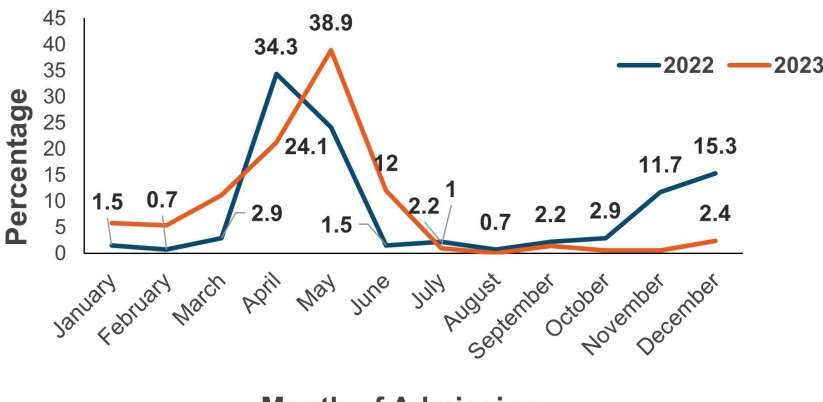

**Fig 1. Monthly Distribution of Admitted Dengue Fever Patients during 2022-2023 in The Muscat Governorate (SQUH and RH).**

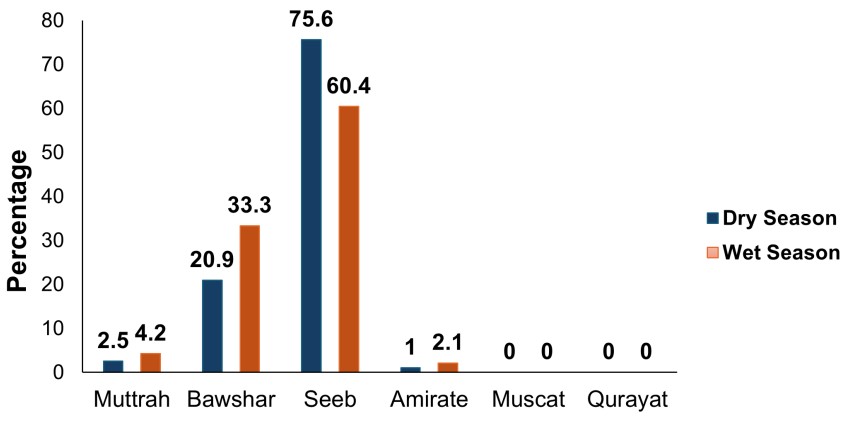

**Fig 2. Distribution of Study Population Across Wilayat in Muscat Governorate Based on Dry vs. Wet Seasons, Oman, from 2022−23.**

A total of 345 patients from Muscat Governorate, the capital city, and the most urbanized region of Oman, were studied, involving two major tertiary healthcare centers: Sultan Qaboos University Hospital (SQUH), located in Seeb, and the Royal Hospital (RH), situated in Bawshar [28]. The locations of Sultan Qaboos University Hospital (SQUH) in Seeb wilayat and the Royal Hospital (RH) in Bawshar wilayat are indicated. Both hospitals served as study sites for the analysis of DF cases admitted during 2022–2023. Both facilities serve as referral centers for the capital and surrounding areas [15], which are characterized by high population density, extensive urban development, and an arid coastal climate that may affect vector ecology and dengue transmission dynamics [29]. In recent years, global warming has contributed to the wider spread of dengue fever and prolonged mosquito activity, resulting in longer transmission seasons [3]. Temperature and rainfall play key roles in mosquito reproduction and virus transmission. Higher temperatures increase dengue virus replication in mosquitoes and shorten the extrinsic incubation period [3]. As observed, a slight surge in DF admissions in these hospitals occurs during the dry season (58%) compared to the wet season (42%). After adjusting for potential confounders, admission to SQUH remained significantly associated with wet-season DF (aOR 1.72; 95% CI 1.06–2.82;

**Table 2. Clinical Characteristics of Patients with Laboratory-Confirmed Dengue Fever by Season, Muscat Governorate, Oman (2022–2023).**

| Characteristics | Total (n=345) | Dry Season[a] (n=201) | Wet Season[b] (n=144) | P-value |
|---|---|---|---|---|
| | n (%) | n (%) | n (%) | |
| **Dengue Severity** | | | | 0.240 |
| Dengue with warning signs | 292 (84.6) | 174 (86.6) | 118 (81.9) | |
| Severe Dengue | 53 (15.4) | 27 (13.4) | 26 (18.1) | |
| **Source** | | | | 0.321 |
| Local | 340 (98.6) | 197 (98.0) | 143 (99.3) | |
| Travel | 5 (1.4) | 4 (2.0) | 1 (0.7) | |
| **Co-infection** | | | | 0.653 |
| Skin Infection | 10 (2.9) | 6 (3.0) | 4 (2.8) | |
| Pneumonia | 5 (1.4) | 4 (2.0) | 1 (0.7) | |
| Bacteremia | 2 (0.6) | 2 (1.0) | 0 (0.0) | |
| Urinary Tract Infection | 3 (0.9) | 2 (1.0) | 1 (0.7) | |
| Chest Infection | 3 (0.9) | 1 (0.5) | 2 (1.4) | |
| **Complications** | | | | 0.651 |
| Neuro Dysfunction | 4 (1.2) | 2 (1.0) | 2 (1.4) | |
| Hepatic Impairment | 9 (2.6) | 4 (2.0) | 5 (3.5) | |
| Platelet Count < 50,000/mm³ | 23 (6.7) | 11 (5.5) | 12 (8.3) | |
| Hemorrhagic Manifestations | 18 (5.2) | 9 (4.5) | 9 (6.3) | |
| Systemic Complication | 8 (2.3) | 6 (3.0) | 2 (1.4) | |
| **Outcome** | | | | 0.807 |
| Discharge Home | 332 (96.2) | 193 (96.0) | 139 (96.5) | |
| Death | 13 (3.8) | 8 (4.0) | 5 (3.5) | |
| **NS1** | 175 (50.7) | 108 (53.7) | 67 (46.5) | 0.407 |
| **IgM** | 85 (24.6) | 53 (26.4) | 32 (22.2) | 0.644 |
| **IgG** | 42 (12.2) | 18 (9.0) | 24 (16.7) | 0.183 |
| **PCR** | 159 (46.1) | 80 (39.8) | 79 (46.1) | **0.006*** |

[a]Dry season: May-October. [b]Wet season: November-April. *Significant at $P \leq 0.05$.

p=0.03), suggesting that referral patterns or catchment areas may influence the observed seasonal distribution rather than inherent disease factors.

The demographic profile of the study population remained consistent throughout the seasons. Adults aged 41–60 years made up the largest group (44.6%), followed by the elderly over 60 years (27.0%), with males slightly outnumbering females (54.8% vs. 45.2%). Most participants were Omani nationals, confirming endogenous transmission. Neither marital status nor employment status varied significantly by season. These findings align with those from regional studies. In Qatar, for example, although no local transmission of dengue fever (DF) has been reported, males aged 20–50 years are more affected by DF than females [30]. Our finding may be due to outdoor occupational exposure and is likely related to behavioral and environmental factors rather than biological susceptibility. A significant portion of cases occurred in older adults; this is similar to a local study, which found that 95% of patients were adults, with an average age of 41 years [6]. This contrasts with endemic countries in Asia, where dengue is more common among children, due to early exposure and community-level immunity [26]. In Oman, where local transmission emerged only recently, the adult dominance reflects a largely unexposed population lacking prior immunity. The study clearly confirmed the endemicity of DF in Muscat Governorate.

**Table 3. Seasonal Distribution of Comorbidities Among Confirmed Dengue Cases in Muscat Governorate, Oman (2022-2023).**

| Comorbidities | Total (n = 345) | Dry Season[a] (n = 201) | Wet Season[b] (n = 144) | P-value |
|---|---|---|---|---|
| Hypertension | 110 (31.9) | 66 (32.8) | 44 (30.6) | 0.654 |
| Diabetic | 87 (25.2) | 43 (21.4) | 44 (30.6) | 0.53 |
| Cancer | 9 (2.6) | 3 (1.5) | 6 (4.2) | 0.124 |
| Chronic Kidney Disease | 22 (6.4) | 15 (7.5) | 7 (4.9) | 0.329 |
| Ischemic Heart Disease | 25 (7.2) | 15 (7.5) | 10 (6.9) | 0.855 |
| Auto immune | 10 (2.9) | 9 (4.5) | 1 (0.7) | **0.039*** |
| Epilepsy | 3 (0.9) | 2 (1.0) | 1 (0.7) | 0.767 |
| Thyroid | 13 (3.8) | 8 (4.9) | 5 (3.5) | 0.807 |
| Asthma | 7 (2.0) | 3 (1.5) | 4 (2.8) | 0.404 |

[a]Dry season: May-October. [b]Wet season: November-April. *Significant at P ≤ 0.05.

**Table 4. Multivariate Logistic Regression Analysis of Factors Associated with Wet-Season Dengue Cases in Muscat Governorate, Oman (2022–2023).**

| Variable | Crude Model | | | Adjusted Model | | |
|---|---|---|---|---|---|---|
| | OR[a] | 95% CI[b] (lower, upper) | P value | aOR[c] | 95% CI (lower, upper) | P value |
| Hospital (SQUH versus RH) | 1.92 | (1.23, 2.99) | 0.004* | 1.72 | (1.06, 2.82) | 0.030* |
| Wilayat | 0.62 | (0.42, 0.92) | 0.016* | 0.78 | (0.51, 1.21) | 0.270 |
| PCR Positivity | 1.45 | (1.11, 1.89) | 0.006* | 1.38 | (1.02, 1.87) | 0.040* |
| Autoimmune | 0.15 | (0.02, 1.19) | 0.073 | 0.15 | (0.02, 1.23) | 0.077 |

[a]OR: Odds Ratio. [b]CI: Confidence Interval. [c]aOR: adjusted Odds Ratio (Multivariable Model). *Significant at P ≤ 0.05.

## Seasonal trends (dry versus wet)

A distinct seasonal pattern was observed in both years 2022 and 2023. In 2022, for example, the admitted DF cases peaked in April (34.3%), followed by another smaller rise in December (15.3%). Similarly, in 2023, the total admitted DF cases were more in May (38.9%). Overall, the concentration of more than 57% of total cases within these two months (April to May) highlights a distinct seasonal pattern, with transmission intensifying in late spring.

Interestingly, another subsequent increase started in November 2022 (11.7%) and continued until May 2023, raising concerns about an unrecorded outbreak event. These months coincide with moderate post-rainfall temperatures and increased humidity, which could enhance mosquito survival and mosquito breeding [30]. Comparable seasonal dynamics have been reported in Saudi Arabia, where a study demonstrated that dengue morbidity and mortality consistently peak during or shortly after the wet season, corresponding with higher rainfall and favorable conditions for Aedes aegypti proliferation [31]. This regional evidence reinforces the importance of climatic variability as a key driver of dengue transmission in the Arabian Peninsula. Similar associations between climate variability and dengue incidence have been documented in previous local study conducted in the Muscat Governorate [6]. Although the total number of dry-season cases was slightly higher, the environmental conditions in some GCC countries, like Qatar, preceding the wet season, likely support vector proliferation [29]. The same pattern was observed before in Oman [6], where the spreading of DF in the Muscat Governorate between 2018–2019 was linked to the proliferation of Aedes aegypti following favorable climatic conditions. Notably, reported admitted DF cases were minimal in July (n = 5), August (n = 1), and September (n = 6), corresponding to the peak summer months in Oman. The extreme heat during this period may limit mosquito survival or reduce human outdoor activity, thereby decreasing potential exposure events. Therefore, intensified vector surveillance, diagnostic preparedness,

increased clinical suspicion and public awareness campaigns should be prioritized in the months leading up to the expected seasonal rise [16].

## Geographic distribution

During 2022–2023 in the Muscat Governorate, most DF admissions were from Seeb Wilayat (69%), followed by Bawshar (26%), with significantly fewer cases elsewhere (p = 0.027). When studying the distribution of cases by seasons (Fig 2), we observed that 75.6% happened in dry seasons compared to 60.4% in wet seasons in Seeb Wilayat. Compared to other Wilayats, more cases happened in wet seasons, raising concerns for increased variability and resistance of DF in Seeb throughout the year. This pattern is consistent with entomological data indicating established Aedes aegypti breeding sites in this district [16]. Seeb and Bowsher's dominance may result from interactions with demographic and environmental factors, such as rapid urbanization, high population density, and water storage practices, which are likely to sustain local transmission [16,18]. Educating about breeding sources is very important, as any container can serve as a breeding ground. People should remove or keep clean those kinds of materials (flower vases, plastic waste that accumulates water, etc.) to prevent outbreaks. Strengthening environmental management and urban planning in high-risk districts is therefore essential to control dengue spread.

## Clinical characteristics

Clinically, most patients presented with dengue with warning signs (84.6%) according to WHO 2009, while severe dengue accounted for 15.4% (Table 2). There were no significant seasonal differences in disease severity (p = 0.24). Most of the admitted DF patients were found to be locally acquired (98.6%), confirming ongoing endemic transmission within Muscat. Hypertension (31.9%) and diabetes mellitus (25.2%) were the most frequent comorbidities, consistent with previous findings [8]. Although these conditions did not vary by season, their presence has been linked in prior studies to a greater risk of complications and poorer outcomes [8]. Laboratory confirmation was primarily based on NS1 antigen and PCR testing, which showed comparable positivity rates between the dry and wet seasons.

In the present study, complications (Table 3) such as thrombocytopenia, hemorrhagic manifestations, and hepatic impairment were infrequent and evenly distributed across seasons. Consistent with previous evidence indicating that dengue infection is primarily associated with specific and rare autoimmune complications, such as autoimmune encephalomyelitis, but not with other autoimmune diseases [32,33], our results showed no significant association between dengue and general autoimmune disorders. In fact, autoimmune disorders were less common among wet-season dengue cases in the crude model (OR = 0.15; 95% CI 0.02–1.19; p = 0.073), although this association was not statistically significant and lost significance after adjustment. Overall mortality was 3.8%, with nearly all patients (96.2%) recovering and being discharged.

The multivariable logistic regression analysis (Table 4) demonstrated that, after adjusting for age, gender, and other covariates, PCR-confirmed diagnosis and hospital of admission were independently associated with wet-season presentation among hospitalized dengue cases in Muscat Governorate. Patients admitted to Sultan Qaboos University Hospital had higher odds of presenting during the wet season compared with those at the Royal Hospital (aOR = 1.72, 95% CI 1.06–2.82, p = 0.030), while individuals who tested positive by PCR were also more likely to be reported in the wet season (aOR = 1.38, 95% CI 1.02–1.87, p = 0.040).

Hospital of admission and PCR positivity were associated with seasonal patterns of hospitalization, reflecting health-system and diagnostic factors rather than epidemiological risk for dengue transmission. PCR can detect dengue virus up to seven days before and after fever onset, making sample timing crucial [34]. Similarly, the association with the hospital of admission may reflect referral patterns and catchment area differences rather than intrinsic disease-related factors. These findings therefore, represent health-system-related associations with wet-season presentation, not direct environmental drivers of dengue transmission.

No significant associations were observed for Wilayat, dengue severity, or autoimmune comorbidity after adjustment, indicating that seasonal presentation was not strongly influenced by individual demographic or clinical characteristics. These findings emphasize several public health priorities. First, proactive vector control and community engagement should be intensified before the onset of the wet season, when conditions favor mosquito breeding development [29,35]. Second, healthcare facilities, particularly SQUH, should ensure adequate preparedness for potential seasonal case surges. Third, patients with underlying chronic comorbidities, such as hypertension and diabetes, warrant closer clinical monitoring due to their higher risk of dengue complications.

## Limitations

This study has several limitations. First, the retrospective design may have been affected by incomplete documentation despite rigorous manual data validation. Second, climatic and entomological variables were not included, limiting direct assessment of environmental drivers of seasonality. Finally, viral serotyping was not available for all patients, restricting strain-specific analysis. Nevertheless, the large sample size and dual-hospital validation enhance the reliability of the findings.

## Conclusion

This study confirms the establishment of locally acquired dengue fever in Muscat Governorate, Oman, with clear seasonal variation and geographic clustering, particularly in Seeb and Bawshar Wilayats. Most hospitalized patients presented with dengue with warning signs and achieved full recovery; however, the occurrence of severe dengue and a 3.8% mortality rate highlights the need for sustained clinical vigilance, especially among patients with chronic comorbidities.

Strengthening vector control interventions, hospital preparedness, and diagnostic capacity prior to the wet season is essential to reduce dengue-related morbidity and mortality. Public health strategies should prioritize community engagement, environmental management, and early case detection. Future studies integrating climatic, entomological, and molecular surveillance data are needed to better characterize transmission dynamics and improve outbreak prediction and response in Oman.

## Acknowledgments

We thank the Department of Disease Surveillance and Control at the Ministry of Health, Oman, for their guidance and support. We appreciate the administration of Sultan Qaboos University Hospital and the Royal Hospital for granting data access. Special thanks go to Ms. Abeer Marhoon Al-Hinai, the Infection Prevention and Control Department at SQUH (notably Mr. Abdullah Al-Rawahi), and the College of Medicine, Department of Family Medicine and Public Health, for their cooperation and assistance throughout this study.

## Declaration of generative AI and AI-assisted technologies in the writing process

In preparing this manuscript, ChatGPT-4 and Grammarly were used as language support tools to assist with grammar, clarity, and sentence structure. ChatGPT-4 was used to help with rephrasing selected parts of the Introduction and Discussion to improve readability, and Grammarly was used for grammatical corrections and general language refinement. All changes suggested by these tools were reviewed and confirmed by the author (IK). The use of these tools was limited to language editing only and did not involve study design, data collection, data analysis, interpretation of results, or the development of scientific content. The scientific content, interpretations, and conclusions of the manuscript are the work of the authors and reflect the original research conducted in this study.

## Author contributions

**Conceptualization:** Ibtisam Khalifa Al-Maskari, Adil Said Al Wahaibi, Sanjay Jaju.

**Data curation:** Ibtisam Khalifa Al-Maskari, Adil Said Al Wahaibi, Hilal Al Sidairi, Sanjay Jaju.

**Formal analysis:** Ibtisam Khalifa Al-Maskari, Hilal Al Sidairi.

**Investigation:** Ibtisam Khalifa Al-Maskari, Adil Said Al Wahaibi, Khaleefathullah A. Sheriff, Hilal Al Sidairi.

**Methodology:** Ibtisam Khalifa Al-Maskari, Zainab M. Al-Zadjali, Amal Malehi, Khaleefathullah A. Sheriff.

**Resources:** Khaleefathullah A. Sheriff, Hilal Al Sidairi.

**Supervision:** Zainab M. Al-Zadjali, Adil Said Al Wahaibi, Sanjay Jaju.

**Validation:** Amal Malehi, Asiya Al Hasni.

**Visualization:** Sanjay Jaju.

**Writing – original draft:** Ibtisam Khalifa Al-Maskari.

**Writing – review & editing:** Zainab M. Al-Zadjali, Amal Malehi, Asiya Al Hasni, Sanjay Jaju.

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
