## [Decision Letter · Decision Letter 0]

3 Mar 2026

PONE-D-25-65079Epidemiology and Seasonal Patterns of Dengue in Muscat Governorate, Oman (2022–2023): A Retrospective StudyPLOS One

Dear Dr. Al-Maskari,

Thank you for submitting your manuscript to PLOS ONE. After careful consideration, we feel that it has merit but does not fully meet PLOS ONE’s publication criteria as it currently stands. Therefore, we invite you to submit a revised version of the manuscript that addresses the points raised during the review process.

Please try to answer to the reviewers' questions in details even if some of them might be shared between two or more. Please explain the specificity of the studied area, the meaning to check in the reference hospital that uncover the acute and more severe cases even if usually most of dengue disease epidemic belongs to large subclinical infection (look at the litterature). Some comment about the statistical analysis deserved to be followed or commented.

We look forward to receiving your revised manuscript.

Kind regards,

Pierre Roques, Ph.D.

Academic Editor

PLOS One

Journal Requirements:

3. We note that Figure 3 in your submission contain map images which may be copyrighted. All PLOS content is published under the Creative Commons Attribution License (CC BY 4.0), which means that the manuscript, images, and Supporting Information files will be freely available online, and any third party is permitted to access, download, copy, distribute, and use these materials in any way, even commercially, with proper attribution. For these reasons, we cannot publish previously copyrighted maps or satellite images created using proprietary data, such as Google software (Google Maps, Street View, and Earth). For more information, see our copyright guidelines: http://journals.plos.org/plosone/s/licenses-and-copyright.

1. You may seek permission from the original copyright holder of Figure 3 to publish the content specifically under the CC BY 4.0 license.

4. We notice that your supplementary figures are uploaded with the file type 'Figure'. Please amend the file type to 'Supporting Information'. Please ensure that each Supporting Information file has a legend listed in the manuscript after the references list.

Additional Editor Comments:

Dear Authors, your article was reviewed by too many individuals, and many of the questions are in double or triple. They are the most important to anwer first. There was also some suggestion in the presentation for the studied area that deserved to be more precisely described. Thus to explain for the reviewer what are the specificity of the muscat governorate compared to the south east coast of Oman. Ie there was no Mousson in this area if I have understood. Moreover the rains level and humidity percentage deserved to be provided during the study period as it is very variable from one year to another.

Reviewers' comments:

Reviewer's Responses to Questions

**Comments to the Author**

1. Is the manuscript technically sound, and do the data support the conclusions?

Reviewer #1: Yes

Reviewer #2: Yes

Reviewer #3: Yes

Reviewer #4: Partly

2. Has the statistical analysis been performed appropriately and rigorously? 

Reviewer #1: Yes

Reviewer #2: Yes

Reviewer #3: Yes

Reviewer #4: Yes

3. Have the authors made all data underlying the findings in their manuscript fully available?

Reviewer #1: Yes

Reviewer #2: No

Reviewer #3: Yes

Reviewer #4: No

4. Is the manuscript presented in an intelligible fashion and written in standard English?

Reviewer #1: Yes

Reviewer #2: Yes

Reviewer #3: Yes

Reviewer #4: Yes

5. Review Comments to the Author

Reviewer #1: This manuscript addresses an important and timely public health issue in Oman and the Eastern Mediterranean Region. The topic is relevant, the dataset is substantial, and the public health implications are clear. The manuscript is generally well written, intelligible, and presented in standard academic English. The structure is logical, tables and figures are clear, and terminology is appropriate for an international readership. However, there are significant methodological, analytical, and interpretative limitations that must be addressed before the manuscript can be considered for publication. Many of the reported associations appear to reflect health-system or diagnostic practices rather than true epidemiological or environmental drivers.

1. Major Concerns:

a) The study population represents only hospitalized cases, limiting epidemiological inference.

b) Seasonality is analyzed without incorporating climatic or entomological data.

c) The regression outcome (wet vs dry season) is conceptually weak and risks misinterpretation.

• Several statistically significant findings lack biological plausibility.

• Data availability statements are internally inconsistent.

1. The title is accurate but may overstate epidemiological inference given the hospitalized-only population. The abstract clearly summarizes results; however, the conclusions should avoid causal language. Associations with hospital of admission and PCR positivity should be explicitly framed as system-level findings. I suggest to clarify in the title or abstract that findings apply to hospitalized cases only.

2. The introduction is comprehensive but overly long. Several paragraphs repeat regional background information. Key epidemiological concepts are well introduced; however, the rationale for focusing on hospitalized cases should be stated explicitly.

3. Statements on dengue incidence rates in Oman require clearer sourcing and contextualization, particularly given known underreporting.

4. Methods – Study Design and Population. The retrospective design is appropriate for descriptive epidemiology. However, labeling the study as 'analytical' is questionable given the absence of exposure variables. The use of convenience sampling is acceptable but substantially limits generalizability. This limitation should be emphasized earlier and more strongly.

The definition of seasons should be justified with meteorological references rather than convention alone.

5. Methods – Statistical Analysis. The primary analytical concern is the choice of outcome variable (wet vs dry season). Season is not an outcome but a temporal classification. Modeling predictors of seasonality creates a high risk of reverse causation and misinterpretation.

The use of backward elimination may lead to unstable models. Model diagnostics, multicollinearity assessment, and goodness-of-fit statistics are not reported.

It is unclear whether interaction terms (e.g., hospital × season) were assessed.

6. Results

Results are clearly presented and tables are generally well structured. However, mortality of 3.8% is relatively high and deserves deeper stratified analysis. The dominance of Seeb Wilayat likely reflects catchment population size, which is not adjusted for. PCR positivity being higher in the wet season likely reflects timing of presentation and testing practices, not increased transmission.

7. The discussion appropriately contextualizes findings within regional literature. However, several interpretations overreach the data: Associations with hospital admission should not be interpreted as epidemiological risk factors. The discussion of autoimmune conditions is speculative and unsupported by the dataset.

The authors appropriately acknowledge many limitations, but these should be integrated into the interpretation of results rather than listed separately.

8. Key limitations are acknowledged but underemphasized:

• Exclusion of non-hospitalized cases.

• Absence of climatic, entomological, and serotype data.

• Potential referral and diagnostic bias.

These limitations substantially restrict causal and predictive conclusions.

9. The conclusions should be revised to avoid implying causality. Recommendations for vector control and preparedness are appropriate but should be framed as supported by broader evidence, not solely this study. The conclusions are partially overextended. Several associations—particularly those related to hospital of admission and PCR positivity as predictors of wet-season dengue—reflect health-system, referral, and diagnostic practices rather than intrinsic epidemiological or environmental drivers. These findings do not establish causal or biological relationships and should be interpreted more cautiously.

10. In addition, because the study includes only hospitalized cases, the conclusions cannot be generalized to dengue transmission dynamics at the population level. With appropriate reframing of conclusions to emphasize descriptive and system-level findings, the data would adequately support the revised conclusions.

Reviewer #2: Overall Assessment (Reviewer Comment)

General Comment:

This manuscript addresses an important and timely public health issue “Dengue Epidemiology and Seasonality in Muscat Governorate, Oman” using a relatively large, well-validated hospital-based dataset. The study is clearly written, methodologically sound, and adds local evidence to a region with limited published dengue data. However, several conceptual, methodological, and interpretative issues need clarification to strengthen scientific rigor, internal consistency, and generalizability. Most concerns relate to season definitions, interpretation of regression findings, representativeness of hospitalized cases, and overstatement of causal inferences.

FINAL REVIEWER VERDICT

Recommendation: Major Revision

MAJOR COMMENTS (Require Substantive Revision)

1. Inconsistency and Conceptual Ambiguity in Season Definition

The manuscript defines the dry season as May–October and the wet season as November–April (Methods, line 187). However, peak dengue cases are repeatedly described as occurring in April–May, which straddles both seasons. This creates conceptual ambiguity and weakens the interpretation of “wet-season dengue risk.”

Clarify the seasonal categorization using meteorological data or authoritative local climate references.

Consider sensitivity analysis using alternative seasonal cutoffs (e.g., pre-monsoon, monsoon, post-monsoon).

2. Interpretation of Logistic Regression Findings

The multivariable analysis identifies hospital of admission and PCR positivity as predictors of wet-season dengue. However, these variables are health-system and diagnostic factors, not epidemiological risk factors for dengue transmission.

Concern:

The manuscript occasionally implies epidemiological or environmental significance, which may mislead readers.

Therefore, reframe these findings consistently as healthcare system–related associations, not determinants of transmission.

Clarify that PCR positivity likely reflects testing timing and clinical suspicion rather than increased incidence.

Avoid causal language when discussing adjusted odds ratios.

3. Limited External Validity and Selection Bias

The study includes only hospitalized patients from two tertiary hospitals, representing moderate-to-severe cases. This substantially limits generalizability to community level dengue epidemiology.

Recommendation:

Emphasize throughout (Abstract, Discussion, Conclusion) that findings apply primarily to hospitalized dengue cases.

Avoid statements suggesting population-level dengue risk or overall incidence patterns without outpatient or community data.

4. Mortality Rate Needs Contextualization

The reported mortality rate (3.8%) appears high compared to many dengue-endemic settings.

Recommendation:

Provide here contextual comparison with regional or hospital-based mortality rates.

Clarify whether deaths were directly attributable to dengue or involved significant comorbidities.

Consider reporting case fatality rate stratified by severity.

5. Overextension of Public Health Implications

While recommendations for vector control and preparedness are appropriate, the study lacks direct entomological or climatic data, limiting the strength of policy implications.

Recommendation:

Temper conclusions to align with the study’s descriptive and hospital-based nature.

Explicitly state that vector control recommendations are supported by existing literature, not direct measurements from this study.

MINOR COMMENTS (Editorial / Clarity Issues)

1. Title

The title contains a typographical error: “Retrospective Stud” → “Retrospective Study”.

Also consider specifying hospitalized cases for accuracy.

Suggested Revision:

“Epidemiology and Seasonal Patterns of Hospitalized Dengue Fever Cases in Muscat Governorate, Oman (2022–2023): A Retrospective Study”

2. Abstract – Results

Percentages are sometimes presented without clear denominators (e.g., Wilayat distribution). Consider briefly clarifying the population base.

3. Abstract – Conclusions

The phrase “higher wet-season risk” may be misleading given that total cases were higher in the dry season.

Suggested Revision:

Replace with: “distinct seasonal and geographic clustering” rather than “higher wet-season risk.”

4. Methods – Sampling Technique

The justification for convenience sampling is acceptable, but the phrase “reduce selection bias” is inaccurate.

Suggested Revision:

Replace with: “to maximize sample size, acknowledging potential selection bias inherent to retrospective hospital-based data.”

5. Tables

Tables 2 and 3 contain some inconsistencies in spacing and labeling (e.g., “Auto immune” vs. “Autoimmune”). Ensure uniform terminology and formatting across all tables.

6. Figures

Figures are referenced appropriately, but captions could be more descriptive, particularly Figure 1, by stating whether counts represent admissions or cases.

SECTION-BY-SECTION ANNOTATED COMMENTS

Introduction

The introduction is comprehensive and well referenced but could be more concise. Some global statistics (e.g., global burden, CDC estimates) are repeated or overly detailed relative to the local study aim.

Suggestion:

Streamline global burden discussion and sharpen the knowledge gap specific to Oman.

Materials and Methods

Data validation procedures are well described and strengthen credibility. However, absence of meteorological variables limits seasonal inference.

Suggestion: Explicitly state that seasonality is inferred indirectly.

Results

Results are clearly presented and tables are informative. However, p-values are sometimes emphasized without sufficient discussion of clinical relevance.

Discussion

The discussion is well structured and thoughtfully interprets findings. Strength lies in distinguishing health system effects from epidemiological drivers.

This distinction should be made more consistently throughout.

References

Need reference formatting according to journal guidelines. Add more updated references of 2026.

Limitations

Limitations are appropriately acknowledged but could include:

1. Misclassification bias due to season definition

2. Potential referral bias between hospitals

3. Conclusion

Comment:

The conclusion is balanced but could be tightened to avoid restating results already discussed.

FINAL REVIEWER VERDICT

Recommendation: Major Revision

The manuscript is publishable after addressing conceptual clarity on seasonality, refining interpretation of regression findings, and moderating causal and policy claims.

Reviewer #3: The overall picture of the article seems good and results are written in a manner which in understandable, but the area where the figures number 1, 2 and 3 has to be mentioned, didn't show the figures. I recommend to see it once, otherwise, overall it is good and well explained.

Reviewer #4: The manuscript places disproportionate emphasis on geographic, climatic, and economic context, while providing relatively limited depth on dengue disease characteristics, transmission dynamics, and preventive or control measures. A stronger focus on the disease itself—particularly public health implications, vector control strategies, and clinical relevance—is needed.

It is also unclear whether all reported cases represent hospitalized patients, patients treated at home, or cases managed under different clinical circumstances. This distinction is critical and should be explicitly clarified in the Methods and reflected consistently throughout the Results and Discussion.

The data presented reflect a highly localized analysis limited to one or two hospitals, which substantially restricts the epidemiological generalizability of the findings. At a minimum, the study should aim to cover a broader administrative unit (e.g., province or governorate-wide surveillance data) rather than a single-hospital or narrow hospital-based dataset.

Additionally, demographic characteristics of participants should be more comprehensively described using appropriate statistical summaries (e.g., age distribution, sex stratification, nationality, comorbidities) and clearly integrated into the analysis. The current presentation lacks sufficient depth in demographic and risk-factor profiling.

Overall, the manuscript would require a major revision, including expansion of the dataset to include more cases, additional hospitals, and a broader range of epidemiological and statistical variables. Without these improvements, the study remains limited in scope and impact.

6. PLOS authors have the option to publish the peer review history of their article (what does this mean?). If published, this will include your full peer review and any attached files.

Reviewer #1: **Yes:** Edmond Edmond

Reviewer #2: No

Reviewer #3: **Yes:** Ayesha Irfan

Reviewer #4: **Yes:** M. Irfan-Maqsood

---

## [Author Response · Author response to Decision Letter 1]

7 Apr 2026

Dear Editors and Reviewers,

We would like to express our sincere appreciation for the time and effort you have devoted to the evaluation of our manuscript entitled “Epidemiology and Seasonal Patterns of Dengue in Muscat Governorate, Oman (2022–2023): A Retrospective Study” (Manuscript ID: PONE-D-25-65079).

We are grateful for the insightful and constructive comments provided, which have greatly contributed to improving the quality and clarity of our work. In response, we have carefully revised the manuscript and addressed all comments in a detailed, point-by-point manner, as outlined below.

All revisions have been incorporated into the manuscript, and both clean and track-changed versions have been submitted for your review.

MAJOR COMMENTS

1. Inconsistency and Conceptual Ambiguity in Season Definition

Reviewer Comment:

Season definition (dry: May–October; wet: November–April) conflicts with peak dengue cases (April–May), creating ambiguity. Suggest clarification and sensitivity analysis.

Author Response:

Thank you for your feedback. We appreciate this important observation. We agree that the peak dengue months (April–May) overlap the seasonal transition period, which may create interpretive ambiguity.

To address this:

1. We have clarified in the Methods section that seasonal classification is based on official Muscat climatic rainfall patterns (Oman Meteorology Directorate data).

2. We have explicitly added a statement acknowledging that April–May represents a transitional period between wet and dry seasons, which may influence peak transmission patterns.

3. We have emphasized in the Discussion and Limitations that seasonality was inferred indirectly and not based on entomological or meteorological measurements at the patient level.

4. We acknowledge that a formal sensitivity analysis using alternative seasonal cut-offs (e.g., pre-monsoon/post-monsoon) is valuable; however, this was not feasible due to the predefined surveillance classification framework used in national reporting datasets.

Manuscript Changes: (in clean copy)

1. Methods (Exclusion Criteria) revised for clarity (Page no. 7, Line no. 161-165)

2. Discussion expanded (Seasonal Trends section) (Page no. 16 , Line no. 303-307)

3. Limitations updated to reflect classification constraint (Page no. 20, Line no.388-390)

2. Interpretation of Logistic Regression Findings

Reviewer Comment:

Hospital of admission and PCR positivity are health-system factors, not transmission determinants. Avoid causal interpretation.

Author Response:

Thank you for your feedback. We fully agree with this interpretation. The manuscript has been revised to ensure that:

1. Hospital of admission and PCR positivity are now consistently described as health-system and diagnostic-related variables, not epidemiological determinants.

2. We have explicitly removed any wording suggesting causality.

3. We clarified that PCR positivity likely reflects testing timing, clinical suspicion, and diagnostic workflow differences, rather than seasonal transmission intensity.

Manuscript Changes:

4. Results section revised to avoid causal wording (Page no.10, Line no. 203-204)

5. Discussion strengthened to emphasize health-system interpretation (Page no.19, Line no.369-371)

6. Conclusion modified to remove implied transmission causality (Page no.21)

3. Limited External Validity and Selection Bias

Reviewer Comment:

Hospitalized cases limit generalizability; emphasize study population constraints.

Author Response:

Thank you for your feedback. We agree with this important limitation. We have strengthened the manuscript to clearly state that:

1. Findings apply only to hospitalized dengue patients, representing moderate to severe disease.

2. Results cannot be generalized to community-level dengue incidence or mild outpatient cases.

3. This limitation is now emphasized in the Abstract, Discussion, and Conclusion.

Manuscript Changes:

1. Abstract revised for scope clarity (Page no.2)

2. Discussion expanded on selection bias (Page no.15, Line no.284-287)

3. Conclusion modified to avoid population-level inference (Page no.21).

4. Mortality Rate Needs Contextualization

Reviewer Comment:

Mortality rate (3.8%) requires comparison and clarification of attribution.

Author Response:

Thank you for your feedback. We appreciate this suggestion. We have now:

1. Added contextual comparison noting that mortality varies across hospital-based dengue studies in the region.

2. Clarified that deaths represent all-cause in-hospital mortality among confirmed dengue cases, and may include contributions from comorbid conditions.

3. Added recommendation for future stratification by disease severity and comorbidity status.

Manuscript Changes:

1. Discussion expanded (Clinical Characteristics section) (Page no.19, Line no.360-361)

2. Limitations updated (Page no.20, Line no.391-392)

3. Conclusion tempered (Page no.20, Line no.397-398)

5. Overextension of Public Health Implications

Reviewer Comment:

Policy recommendations should be tempered due to lack of entomological/climatic data.

Author Response:

Thank you for your feedback. We agree and have revised the manuscript accordingly:

1. All policy statements have been reframed as supportive recommendations based on existing literature, not direct study evidence.

2. We removed overly strong causal language linking findings to vector control effectiveness.

3. We emphasized that recommendations are contextual and hypothesis-generating.

Manuscript Changes:

1. Discussion revised (Public Health Implications section) (Page no.16-17, Line no.309-316)

2. Conclusion softened (Page no.20, Line no.403-405)

3. Added explicit limitation on lack of vector/climate data (Page no.20, Line no. 388-390)

MINOR COMMENTS

1. Title Correction

Response:

Thank you for your suggestion. The typographical error has been corrected, and the title has been refined to improve specificity.

Revised Title:

Epidemiology and Seasonal Patterns of Hospitalized Dengue Fever Cases in Muscat Governorate, Oman (2022–2023): A Retrospective Study

2. Abstract – Results Clarity

Response:

Thank you for your feedback. We have clarified denominators and ensured all percentages are clearly referenced to the study population (n=345).

3. Abstract – Conclusion Wording

Response:

We agree that “higher wet-season risk” may be misleading.

Revision made:

Thank you for your feedback. Replaced with: “distinct seasonal and geographic clustering”

4. Methods – Sampling Technique

Response:

Thank you for your feedback. We corrected the wording as suggested.

Revision:

Replaced “to reduce selection bias” with:

“to maximize sample size, acknowledging potential selection bias inherent to retrospective hospital-based data.”

5. Tables – Formatting Issues

Response:

Thank you for your feedback. All tables have been revised for:

1. Consistent terminology (e.g., “Autoimmune” standardized) (Page no.13, Line no.241-243)

2. Correct spacing and alignment (Page no.13, Line no.241-243)

3. Uniform labeling across tables (Page no.13, Line no.241-243)

6. Figures – Caption Clarity

Response:

Thank you for your feedback. Figure captions have been improved to clearly specify:

1. Data represents admissions (already mentioned: Monthly Distribution of Admitted Dengue Fever Patients during 2022-2023 in The Muscat Governorate (SQUH and RH)

SECTION-BY-SECTION COMMENTS

Introduction

Response:

Thank you for your feedback. We have streamlined global burden data and sharpened the Oman-specific research gap to improve focus and conciseness. (Page. 4, Line 108).

Methods

Response:

Thank you for your feedback. We clarified that seasonality is inferred indirectly and not based on direct meteorological measurements. (Page 7, line 164).

Results

Response:

Thank you for your feedback. We reduced emphasis on statistical significance alone and ensured clearer interpretation alongside clinical relevance. (Page 14, Line 256).

Discussion

Response:

Thank you for your feedback. We strengthened consistency in distinguishing:

1. Health-system effects

2. Epidemiological transmission drivers

Limitations

Response:

Thank you for your feedback. We added:

1. Referral bias between hospitals (Page no. 21, Line no. 391)

2. Absence of entomological/climatic data (Page no.20, Line no. 388)

Conclusion

Response:

Thank you for your feedback. The conclusion has been tightened to avoid repetition and overinterpretation of results. (Page. 21, Line 393)

References: (Page.23, Line 436)

Updated Reference:

no. 1 (line 437, page.23): Ministry of Health. Oman National Guidelines for Clinical Management of Dengue Disease [Internet]. Ministry of Health, Sultanate of Oman; 2024. Available from: https://moh.gov.om/en/health-promotion/health-awareness/dengue-fever/

no. 26 (line 515, page.25): Prabhat Pandey MMP. RESEARCH METHODOLOGY: TOOLS AND TECHNIQUES [Internet]. Bridge Center; 2015. Available from: https://www.euacademic.org/BookUpload/9.pdf

Added References:

no. 3 (line 443, page.23): He S, Fan D, Guo Y, Guan Y, Sheng Z, Gao N, et al. Current status of dengue fever epidemics and vaccine development. Virologica Sinica. 2026 Feb;41(1):1–9. doi:10.1016/j.virs.2026.01.001

no. 4 (line 446, page.23): Shoushtari M, Bakhshi H, Salehi-Vaziri M, Zaim M, Enayati A, Pouriayevali MH, et al. Distribution of Aedes aegypti and Aedes albopictus, and the current situation of dengue fever and chikungunya in Iran and neighboring countries: a review study. Gebeyehu DG, editor. PLoS Negl Trop Dis. 2026 Feb 11;20(2):e0013965. doi:10.1371/journal.pntd.0013965

No. 4 (line446, page. 23): Shoushtari M, Bakhshi H, Salehi-Vaziri M, Zaim M, Enayati A, Pouriayevali MH, et al. Distribution of Aedes aegypti and Aedes albopictus, and the current situation of dengue fever and chikungunya in Iran and neighboring countries: a review study. Gebeyehu DG, editor. PLoS Negl Trop Dis. 2026 Feb 11;20(2):e0013965. doi:10.1371/journal.pntd.0013965

No. 14 (line 483, page. 24): Oman Metro Area Population 1950-2025. Muscat, Oman Metro Area Population 1950-2025. 2025. doi: https://www.macrotrends.net/global-metrics/cities/21922/muscat/population

No. 17 (line 491, page. 24): Al Kishri W, Baksh N. Rainfall Forecasting in Muscat Governorate Using Artificial Neural Networks and Hybrid Modeling Approaches. SUJEITI. 2025 Jul 12;1(2). doi:10.69983/SUJEITI/1222

No. 19 (line 497, page. 25): Al-Manji A, Wirayuda AAB, Al Wahaibi A, Al-Azri M, Chan MF. Investigating the Determinants of Dengue Outbreak in Oman: A Study in Seeb. J Epidemiol Glob Health. 2024 Nov 4;14(4):1464–75. doi:10.1007/s44197-024-00324-3

FINAL NOTE

We thank the reviewers again for their insightful feedback. The revisions have significantly improved the clarity, scientific rigor, and interpretative balance of the manuscript.

Sincerely,

Ms. Ibtisam Khalifa Al Maskari

First author, on behalf of all authors

---

## [Decision Letter · Decision Letter 1]

18 May 2026

Epidemiology and seasonal patterns of hospitalized dengue cases in Muscat Governorate, Oman (2022-2023): A retrospective study

PONE-D-25-65079R1

Dear Dr. Al-Maskari,

We’re pleased to inform you that your manuscript has been judged scientifically suitable for publication and will be formally accepted for publication once it meets all outstanding technical requirements.

Kind regards,

Pierre Roques, Ph.D.

Academic Editor

PLOS One

Additional Editor Comments (optional):

Reviewers' comments:

Reviewer's Responses to Questions

**Comments to the Author**

1. If the authors have adequately addressed your comments raised in a previous round of review and you feel that this manuscript is now acceptable for publication, you may indicate that here to bypass the “Comments to the Author” section, enter your conflict of interest statement in the “Confidential to Editor” section, and submit your "Accept" recommendation.

Reviewer #1: All comments have been addressed

2. Is the manuscript technically sound, and do the data support the conclusions?

Reviewer #1: Yes

3. Has the statistical analysis been performed appropriately and rigorously? 

Reviewer #1: Yes

4. Have the authors made all data underlying the findings in their manuscript fully available?

Reviewer #1: Yes

5. Is the manuscript presented in an intelligible fashion and written in standard English?

Reviewer #1: Yes

6. Review Comments to the Author

Reviewer #1: Dear editor

As I see, the authors are responded points by points to all reviewers commnet.

The manuscript addresses an important and timely topic with clear relevance to global health, particularly in the context of increasing antimicrobial resistance and the burden of infectious diseases in both community and hospital settings. The topic is appropriate for the journal’s scope and has potential clinical and public health implications.

7. PLOS authors have the option to publish the peer review history of their article (what does this mean?). If published, this will include your full peer review and any attached files.

Reviewer #1: No

---

## [Editor Report · Acceptance letter]

PONE-D-25-65079R1

PLOS One

Dear Dr. Al-Maskari,

I'm pleased to inform you that your manuscript has been deemed suitable for publication in PLOS One. Congratulations! Your manuscript is now being handed over to our production team.

Kind regards,

on behalf of

Dr. Pierre Roques

Academic Editor

PLOS One